# Toward Understanding the Impact of Staleness in Distributed Machine Learning

**Wei Dai,**[*] **Yi Zhou**[†]**, Nanqing Dong**[§]**, Hao Zhang**[§]**, Eric P. Xing**[§]
[*]Apple Inc, [†]Duke University, and [§]Petuum Inc
daviddai@apple.com, yi.zhou610@duke.edu
{nanqing.dong, hao.zhang, eric.xing}@petuum.com

## Abstract

Most distributed machine learning (ML) systems store a copy of the model parameters locally on each machine to minimize network communication. In practice, in order to reduce synchronization waiting time, these copies of the model are not necessarily updated in lock-step, and can become *stale*. Despite much development in large-scale ML, the effect of staleness on the learning efficiency is inconclusive, mainly because it is challenging to control or monitor the staleness in complex distributed environments. In this work, we study the convergence behaviors of a wide array of ML models and algorithms under delayed updates. Our extensive experiments reveal the rich diversity of the effects of staleness on the convergence of ML algorithms and offer insights into seemingly contradictory reports in the literature. The empirical findings also inspire a new convergence analysis of SGD in non-convex optimization under staleness, matching the best-known convergence rate of $\mathcal{O}(1/\sqrt{T})$.

## 1 Introduction

With the advent of big data and complex models, there is a growing body of works on scaling machine learning under synchronous and non-synchronous[1] distributed execution (Dean et al., 2012; Goyal et al., 2017; Li et al., 2014a). These works, however, point to seemingly contradictory conclusions on whether non-synchronous execution outperforms synchronous counterparts in terms of absolute convergence, which is measured by the wall clock time to reach the desired model quality. For deep neural networks, Chilimbi et al. (2014); Dean et al. (2012) show that fully asynchronous systems achieve high scalability and model quality, but others argue that synchronous training converges faster (Chen et al., 2016; Cui et al., 2016). The disagreement goes beyond deep learning models: Ho et al. (2013); Zhang & Kwok (2014); Langford et al. (2009); Lian et al. (2015); Recht et al. (2011) empirically and theoretically show that many algorithms scale effectively under non-synchronous settings, but McMahan & Streeter (2014); Mitliagkas et al. (2016); Hadjis et al. (2016) demonstrate significant penalties from asynchrony.

The crux of the disagreement lies in the trade-off between two factors contributing to the absolute convergence: *statistical efficiency* and *system throughput*. Statistical efficiency measures convergence per algorithmic step (e.g., a mini-batch), while system throughput captures the performance of the underlying implementation and hardware. Non-synchronous execution can improve system throughput due to lower synchronization overheads, which is well understood (Ho et al., 2013; Chen et al., 2016; Cui et al., 2014; Chilimbi et al., 2014; Dai et al., 2015). However, by allowing various workers to use *stale* versions of the model that do not always reflect the latest updates, non-synchronous systems can exhibit lower statistical efficiency (Chen et al., 2016; Cui et al., 2016). How statistical efficiency and system throughput trade off in distributed systems, however, is far from clear.

The difficulties in understanding the trade-off arise because statistical efficiency and system throughput are coupled during execution in distributed environments. Non-synchronous executions are in general non-deterministic, which can be difficult to profile. Furthermore, large-scale experiments

---

[*]The work is conducted at Petuum Inc.
[1]We use the term "non-synchronous" to include both fully asynchronous model (Recht et al., 2011) and bounded asynchronous models such as Stale Synchronous Parallel (Ho et al., 2013).

are sensitive to the underlying hardware and software artifacts, which confounds the comparison between studies. Even when they are controlled, innocuous change in the system configurations such as adding more machines or sharing resources with other workloads can inadvertently alter the underlying staleness levels experienced by ML algorithms, masking the true effects of staleness.

Understanding the impact of staleness on ML convergence independently from the underlying distributed systems is a crucial step towards decoupling statistical efficiency from the system complexity. The gleaned insights can also guide distributed ML system development, potentially using different synchronization for different problems. In particular, we are interested in the following aspects: Do ML algorithms converge under staleness? To what extent does staleness impact the convergence?

By resorting to simulation study, we side step the challenges faced in distributed execution. We study the impact of staleness on a diverse set of models: Convolutional Neural Networks (CNNs), recurrent neural networks (RNNs), Deep Neural Networks (DNNs), multi-class Logistic Regression (MLR), Matrix Factorization (MF), Latent Dirichlet Allocation (LDA), and Variational Autoencoders (VAEs). They are addressed by 7 algorithms, spanning across optimization, sampling, and blackbox variational inference. Our findings suggest that while some algorithms are more robust to staleness, no ML method is immune to the negative impact of staleness. We find that all investigated algorithms reach the target model quality under moderate levels of staleness, but the convergence can progress very slowly or fail under high staleness levels. The effects of staleness are also problem dependent. For CNNs, DNNs, and RNNs, the staleness slows down deeper models more than shallower counterparts. For MLR, a convex objective, staleness has minimal effect. Different algorithms respond to staleness very differently. For example, high staleness levels incur more statistical penalty for Momentum methods than stochastic gradient descent (SGD) and Adagrad (Duchi et al., 2011). Separately, Gibbs sampling for LDA is highly resistant to staleness up to a certain level, beyond which it does not converge to a fixed point. Overall, it appears that staleness is a key governing parameter of ML convergence.

To gain deeper insights, for gradient-based methods we further introduce *gradient coherence* along the optimization path, and show that gradient coherence is a possible explanation for an algorithm's sensitivity to staleness. In particular, our theoretical result establishes the $\mathcal{O}(1/\sqrt{T})$ convergence rate of the asynchronous SGD in nonconvex optimization by exploiting gradient coherence, matching the rate of best-known results (Lian et al., 2015).

## 2    RELATED WORK

Staleness is reported to help absolute convergence for distributed deep learning in Chilimbi et al. (2014); Dean et al. (2012); Xing et al. (2015) and has minimal impact on convergence (Mitliagkas et al., 2016; Hadjis et al., 2016; Lian et al., 2015; Dai et al., 2013; Zhou et al., 2018; 2016). But Chen et al. (2016); Cui et al. (2016) show significant negative effects of staleness. LDA training is generally insensitive to staleness (Smola & Narayanamurthy, 2010; Yuan et al., 2015; Wei et al., 2015; Ho et al., 2013), and so is MF training (Yun et al., 2013; Low et al., 2012; Cui et al., 2014; Zhang & Kwok, 2014). However, none of their evaluations quantifies the level of staleness in the systems. By explicitly controlling the staleness, we decouple the distributed execution, which is hard to control, from ML convergence outcomes.

We focus on algorithms that are commonly used in large-scale optimization (Goyal et al., 2017; Chen et al., 2016; Dean et al., 2012), instead of methods specifically designed to minimize synchronization (Neiswanger et al., 2013; Scott et al., 2016; Jordan et al., 2013). Non-synchronous execution has theoretical underpinning (Li et al., 2014b; Ho et al., 2013; Zhang & Kwok, 2014; Lian et al., 2015; Recht et al., 2011). Here we study algorithms that do not necessarily satisfy assumptions in their analyses.

## 3    METHODS

We study six ML models and focus on algorithms that lend itself to data parallelism, which a primary approach for distributed ML. Our algorithms span optimization, sampling, and black box variational inference. Table 1 summarizes the studied models and algorithms.

**Simulation Model.** Each update generated by worker $p$ needs to be propagated to both worker $p$'s model cache and other worker's model cache. We apply a uniformly random delay model to these updates that are in transit. Specifically, let $u_p^t$ be the update generated at iteration $t$ by worker $p$. For

each worker $p'$ (including $p$ itself), our delay model applies a delay $r_{p,p'}^t \sim \text{Categorical}(0, 1, .., s)$, where $s$ is the maximum delay and $\text{Categorical}()$ is the categorical distribution placing equal weights on each integer[2]. Under this delay model, update $u_p^t$ shall arrive at worker $p'$ at the start of iteration $t + 1 + r_{p,p'}^t$. The average delay under this model is $\frac{1}{2}s + 1$. Notice that for one worker with $s = 0$ we reduce to the sequential setting. Since model caches on each worker are symmetric, we use the first worker's model to evaluate the model quality. Finally, we are most interested in measuring convergence against the logical time, and wall clock time is in general immaterial as the simulation on a single machine is not optimized for performance.

## 3.1 MODELS AND ALGORITHMS

| Model | Algorithms | Key Parameters | Dataset |
|---|---|---|---|
| CNN RNN | SGD | $\eta$ | CIFAR10 (CNN) Penn Treebank (RNN) |
| | Momentum SGD | $\eta$, momentum=0.9 | |
| | Adam | $\eta, \beta_1 = 0.9, \beta_2 = 0.999$ | |
| | Adagrad | $\eta$ | |
| | RMSProp | $\eta$, decay=0.9, momentum=0 | |
| DNN/MLR | SGD | $\eta = 0.01$ | MNIST |
| | Adam | $\eta = 0.001, \beta_1 = 0.9, \beta_2 = 0.999$ | |
| LDA | Gibbs Sampling | $\alpha = 0.1, \beta = 0.1$ | 20 NewsGroup |
| MF | SGD | $\eta = 0.005$, rank=5, $\lambda = 0.0001$ | MovieLens1M |
| VAE | Blackbox VI (SGD, Adam) | Optimization parameters same as MLR/DNN | MNIST |

Table 1: Overview of the models, algorithms (Qian, 1999; Duchi et al., 2011; Kingma & Ba, 2014; Hinton, 2012; Griffiths & Steyvers, 2004), and dataset (Krizhevsky & Hinton, 2009; Marcus et al., 1993; LeCun, 1998; Harper & Konstan, 2016; Rennie) in our study. $\eta$ denotes learning rate, which, if not specified, are tuned empirically for each algorithm and staleness level, $\beta_1, \beta_2$ are optimization hyperparameters (using common default values). $\alpha, \beta$ in LDA are Dirichlet priors for document topic and word topic random variables, respectively.

**Convolutional Neural Networks (CNNs)** have been a strong focus of large-scale training, both under synchronous (Goyal et al., 2017; Cui et al., 2016; Coates et al., 2013) and non-synchronous (Chilimbi et al., 2014; Dean et al., 2012; Chen et al., 2016; Hadjis et al., 2016) training. We consider residual networks with $6n + 2$ weight layers (He et al., 2016). The networks consist of 3 groups of $n$ residual blocks, with 16, 32, and 64 feature maps in each group, respectively, followed by a global pooling layer and a softmax layer. The residual blocks have the same construction as in (He et al., 2016). We measure the model quality using test accuracy. For simplicity, we omit data augmentation in our experiments.

**Deep Neural Networks (DNNs)** are neural networks composed of fully connected layers. Our DNNs have 1 to 6 hidden layers, with 256 neurons in each layer, followed by a softmax layer. We use rectified linear units (ReLU) for nonlinearity after each hidden layer (Nair & Hinton, 2010). **Multi-class Logistic Regression (MLR)** is the special case of DNN with 0 hidden layers. We measure the model quality using test accuracy.

**Matrix factorization (MF)** is commonly used in recommender systems and have been implemented at scale (Yun et al., 2013; Low et al., 2012; Cui et al., 2014; Zhang & Kwok, 2014; Kim et al., 2016; Ho et al., 2013; Kumar et al., 2014). Let $D \in \mathbb{R}^{M \times N}$ be a partially filled matrix, MF factorizes $D$ into two factor matrices $L \in \mathbb{R}^{M \times r}$ and $R \in \mathbb{R}^{N \times r}$ ($r \ll \min(M, N)$ is the user-defined rank). The $\ell_2$-penalized optimization problem is: $\min_{L,R} \frac{1}{|D_{obs}|} \left\{ \sum_{(i,j) \in D_{obs}} ||D_{ij} - \sum_{k=1}^K L_{ik} R_{kj}||^2 + \lambda(||L||_F^2 + ||R||_F^2) \right\}$ where $|| \cdot ||_F$ is the Frobenius norm and $\lambda$ is the regularization parameter. We partition observations $D$ to workers while treating $L, R$ as shared model parameters. We optimize MF via SGD, and measure model quality by training loss defined by the objective function above.

**Latent Dirichlet Allocation (LDA)** is an unsupervised method to uncover hidden semantics ("topics") from a group of documents, each represented as a bag of tokens. LDA has been scaled under non-synchronous execution (Ahmed et al., 2012; Low et al., 2012; Yuan et al., 2015) with great success. Further details are provided in Appendix.

---

[2]We find that geometrically distributed delays, presented in the sequel, have qualitatively similar impacts on convergence. We defer read-my-write consistency to future work.

**Variational Autoencoder (VAE)** is commonly optimized by black box variational inference, which can be considered as a hybrid of optimization and sampling methods. The inputs to VAE training include two sources of stochasticity: the data sampling $x$ and samples of random variable $\epsilon$. We measure the model quality by test loss. We use DNNs with 1~3 layers as the encoders and decoders in VAE, in which each layer has 256 units furnished with rectified linear function for non-linearity. The model quality is measured by the training objective value, assuming continuous input $x$ and isotropic Gaussian prior $p(z) \sim \mathcal{N}(0, I)$.

## 4 EXPERIMENTS

We use batch size 32 for CNNs, DNNs, MLR, and VAEs[34]. For MF, we use batch size of 25000 samples, which is 2.5% of the MovieLens dataset (1M samples). We study staleness up to $s = 50$ on 8 workers, which means model caches can miss updates up to 8.75 data passes. For LDA we use $\frac{D}{10P}$ as the batch size, where $D$ is the number of documents and $P$ is the number of workers. We study staleness up to $s = 20$, which means model caches can miss updates up to 2 data passes. We measure time in terms of the amount of work performed, such as the number of batches processed.

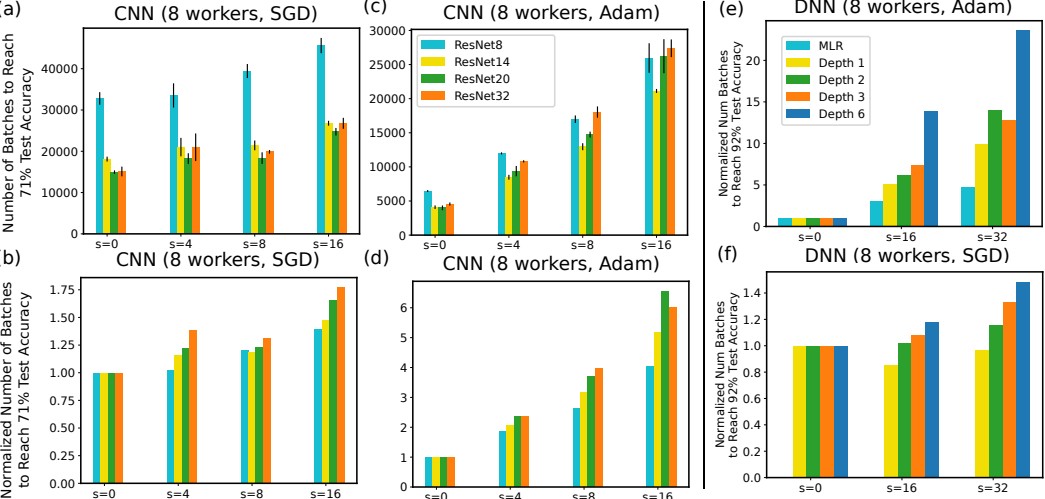

Figure 1: **(a)(c)** The number of batches to reach 71% test accuracy on CIFAR10 for 4 variants of ResNet with varying staleness, using 8 workers and SGD (learning rate 0.01) and Adam (learning rate 0.001). The mean and standard deviation are calculated over 3 randomized runs. **(b)(d)** The same metrics as (a)(c), but each model is normalized by the value under staleness 0 ($s = 0$), respectively. **(e)(f)** The number of batches to reach 92% accuracy for MLR and DNN with varying depths, normalized by the value under staleness 0. MLR with SGD does not converge within the experiment horizon (77824 batches) and thus is omitted in (f).

**Convergence Slowdown.** Perhaps the most prominent effect of staleness on ML algorithms is the slowdown in convergence, evident throughout the experiments. Fig. 1 shows the number of batches needed to reach the desired model quality for CNNs and DNNs/MLR with varying network depths and different staleness ($s = 0, ..., 16$). Fig. 1(b)(d) show that convergence under higher level of staleness requires more batches to be processed in order to reach the same model quality. This additional work can potentially be quite substantial, such as in Fig. 1(d) where it takes up to 6x more batches compared with settings without staleness ($s = 0$). It is also worth pointing out that while there can be a substantial slowdown in convergence, the optimization still reaches desirable models under most cases in our experiments. When staleness is geometrically distributed (Fig. 4(c)), we observe similar patterns of convergence slowdown.

We are not aware of any prior work reporting slowdown as high as observed here. This finding has important ramifications for distributed ML. Usually, the moderate amount of workload increases due to parallelization errors can be compensated by the additional computation resources and higher system throughput in the distributed execution. However, it may be difficult to justify spending large

---

[3] Non-synchronous execution allows us to use small batch sizes, eschewing the potential generalization problem with large batch SGD (Keskar et al., 2016; Masters & Luschi, 2018).

[4] We present RNN results in the Appendix.

amount of resources for a distributed implementation if the statistical penalty is too high, which should be avoided (e.g., by staleness minimization system designs or synchronous execution).

**Model Complexity.** Fig. 1 also reveals that the impact of staleness can depend on ML parameters, such as the depths of the networks. Overall we observe that staleness impacts deeper networks more than shallower ones. This holds true for SGD, Adam, Momentum, RMSProp, Adagrad (Fig. 1), and other optimization schemes, and generalizes to other numbers of workers (see Appendix)[5].

This is perhaps not surprising, given the fact that deeper models pose more optimization challenges even under the sequential settings (Glorot & Bengio, 2010; He et al., 2016), though we point out that existing literature does not explicitly consider model complexity as a factor in distributed ML (Lian et al., 2015; Goyal et al., 2017). Our results suggest that the staleness level acceptable in distributed training can depend strongly on the complexity of the model. For sufficiently complex models it may be more advantageous to eliminate staleness altogether and use synchronous training.

**Algorithms' Sensitivity to Staleness.** Staleness has uneven impacts on different SGD variants. Fig. 2 shows the amount of work (measured in the number of batches) to reach the desired model quality for five SGD variants. Fig. 2(d)(e)(f) reveals that while staleness generally increases the number of batches needed to reach the target test accuracy, the increase can be higher for certain algorithms, such as Momentum. On the other hand, Adagrad appear to be robust to staleness[6]. Our finding is consistent with the fact that, to our knowledge, all existing successful cases applying non-synchronous training to deep neural networks use SGD (Dean et al., 2012; Chilimbi et al., 2014). In contrast, works reporting subpar performance from non-synchronous training often use momentum, such as RMSProp with momentum (Chen et al., 2016) and momentum (Cui et al., 2016). Our results suggest that these different outcomes may be partly driven by the choice of optimization algorithms, leading to the seemingly contradictory reports of whether non-synchronous execution is advantageous over synchronous ones.

**Effects of More Workers.** The impact of staleness is amplified by the number of workers. In the case of MF, Fig. 3(b) shows that the convergence slowdown in terms of the number of batches (normalized by the convergence for $s = 0$) on 8 workers is more than twice of the slowdown on 4 workers. For example, in Fig. 3(b) the slowdown at $s = 15$ is ~3.4, but the slowdown at the same staleness level on 8 workers is ~8.2. Similar observations can be made for CNNs (Fig. 3). This can be explained by the fact that additional workers amplifies the effect of staleness by (1) generating updates that will be subject to delays, and (2) missing updates from other workers that are subject to delays.

**LDA.** Fig. 3(c)(d) show the convergence curves of LDA with different staleness levels for two settings varying on the number of workers and topics. Unlike the convergence curves for SGD-based algorithms (see Appendix), the convergence curves of Gibbs sampling are highly smooth, even under high staleness and a large number of workers. This can be attributed to the structure of log likelihood objective function (Griffiths & Steyvers, 2004). Since in each sampling step we only update the count statistics based on a portion of the corpus, the objective value will generally change smoothly.

Staleness levels under a certain threshold ($s \leq 10$) lead to convergence, following indistinguishable log likelihood trajectories, regardless of the number of topics ($K = 10, 100$) or the number of workers (2–16 workers, see Appendix). Also, there is very minimal variance in those trajectories. However, for staleness beyond a certain level ($s \geq 15$), Gibbs sampling does not converge to a fixed point. The convergence trajectories are distinct and are sensitive to the number of topics and the number of workers. There appears to be a "phase transition" at a certain staleness level that creates two distinct phases of convergence behaviors[7]. We believe this is the first report of a staleness-induced failure case for LDA Gibbs sampling.

**VAE** In Fig. 3(e)(f), VAEs exhibit a much higher sensitivity to staleness compared with DNNs (Fig. 1(e)(f)). This is the case even considering that VAE with depth 3 has 6 weight layers, which

---

[5]ResNet8 takes more batches to reach the same model quality than deeper networks in Fig. 1(a) because, with SGD, ResNet8's final test accuracy is about 73% in our setting, while ResNet20's final test accuracy is close to 75%. Therefore, deeper ResNet can reach the same model accuracy in the earlier part of the optimization path, resulting in lower number of batches in Fig. 1(a). However, when the convergence time is normalized by the non-stale (s=0) value in Fig. 1(b), we observe the impact of staleness is higher on deeper models.

[6] Many synchronous systems uses batch size linear in the number of workers (e.g., (Goyal et al., 2017)). We preserve the same batch size and more workers simply makes more updates in each iteration.

[7]We leave the investigation into this distinct phenomenon as future work.

has a comparable number of model parameters and network architecture to DNNs with 6 layers. We hypothesize that this is caused by the additional source of stochasticity from the sampling procedure, in addition to the data sampling process.

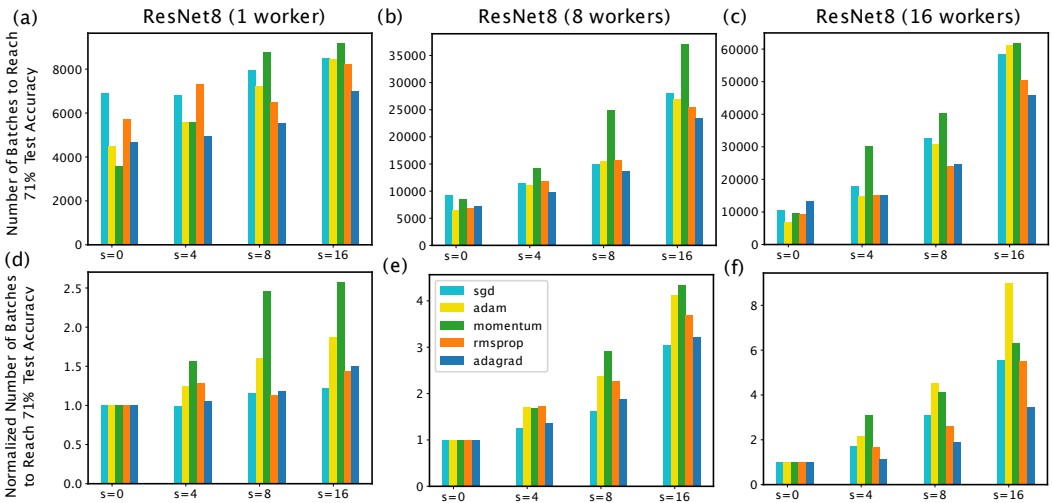

Figure 2: (a)(b)(c) The number of batches to reach 71% test accuracy on 1, 8, 16 workers with staleness $s = 0, ..., 16$ using ResNet8. We consider 5 variants of SGD: SGD, Adam, Momentum, RMSProp, and Adagrad. For each staleness level, algorithm, and the number of workers, we choose the learning rate with the fastest time to 71% accuracy from $\{0.001, 0.01, 0.1\}$. (d)(e)(f) show the same metric but each algorithm is normalized by the value under staleness 0 ($s = 0$), respectively, with possibly different learning rate.

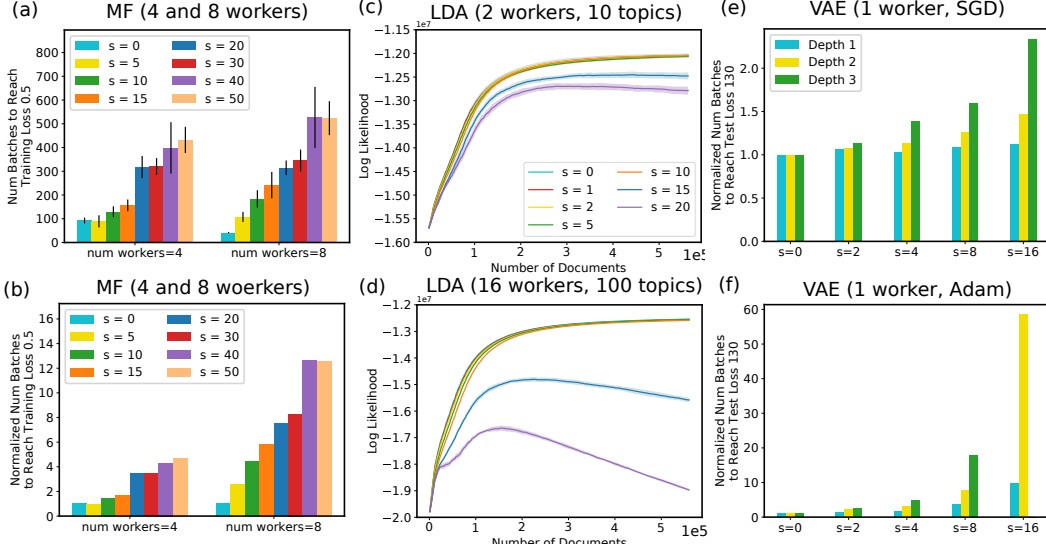

Figure 3: (a) The number of batches to reach training loss of 0.5 for Matrix Factorization (MF). (b) shows the same metric in (a) but normalized by the values of staleness 0 of each worker setting, respectively (4 and 8 workers). (c)(d) Convergence of LDA log likelihood using 10 and 100 topics under staleness levels $s = 0, ..., 20$, with 2 and 16 workers. The convergence is recorded against the number of documents processed by Gibbs sampling. The shaded regions are 1 standard deviation around the means (solid lines) based on 5 randomized runs. (e)(f) The number of batches to reach test loss 130 by Variational Autoencoders (VAEs) on 1 worker, under staleness $s = 0, ..., 16$. We consider VAEs with depth 1, 2, and 3 (the number of layers in the encoder and the decoder networks, separately). The numbers of batches are normalized by $s = 0$ for each VAE depth, respectively. Configurations that do not converge to the desired test loss are omitted in the graph, such as Adam optimization for VAE with depth 3 and $s = 16$.

## 5 GRADIENT COHERENCE AND CONVERGENCE OF ASYNCHRONOUS SGD

We now provide theoretical insight into the effect of staleness on the observed convergence slowdown. We focus on the challenging asynchronous SGD (Async-SGD) case, which characterizes the neural

network models, among others. Consider the following nonconvex optimization problem

$$\min_{\mathbf{x} \in \mathbb{R}^d} F(\mathbf{x}) := \frac{1}{n} \sum_{i=1}^{n} f_i(\mathbf{x}), \tag{P}$$

where $f_i$ corresponds to the loss on the $i$-th data sample, and the objective function is assumed to satisfy the following standard conditions:

**Assumption 1.** *The objective function $F$ in the problem (P) satisfies:*

1. *Function $F$ is continuously differentiable and bounded below, i.e., $\inf_{\mathbf{x} \in \mathbb{R}^d} F(\mathbf{x}) > -\infty$;*
2. *The gradient of $F$ is $L$-Lipschitz continuous.*

Notice that we allow $F$ to be nonconvex. We apply the Async-SGD to solve the problem (P). Let $\xi(k)$ be the mini-batch of data indices sampled from $\{1, \dots, n\}$ uniformly at random by the algorithm at iteration $k$, and $|\xi(k)|$ is the mini-batch size. Denote mini-batch gradient as $\nabla f_{\xi(k)}(\mathbf{x}_k) := \sum_{i \in \xi(k)} \nabla f_i(\mathbf{x}_k)$. Then, the update rule of Async-SGD can be written as

$$\mathbf{x}_{k+1} = \mathbf{x}_k - \frac{\eta_k}{|\xi(\tau_k)|} \nabla f_{\xi(\tau_k)}(\mathbf{x}_{\tau_k}), \tag{Async-SGD}$$

where $\eta_k$ corresponds to the stepsize, $\tau_k$ denotes the delayed clock and the maximum staleness is assumed to be bounded by $s$. This implies that $k - s + 1 \leq \tau_k \leq k$.

The optimization dynamics of Async-SGD is complex due to the nonconvexity and the uncertainty of the delayed updates. Interestingly, we find that the following notion of gradient coherence provides insights toward understanding the convergence property of Async-SGD.

**Definition 1** (Gradient coherence). *The gradient coherence at iteration $k$ is defined as*

$$\mu_k := \min_{k-s+1 \leq t \leq k} \frac{\langle \nabla F(\mathbf{x}_k), \nabla F(\mathbf{x}_t) \rangle}{\|\nabla F(\mathbf{x}_k)\|^2}.$$

Parameter $\mu_k$ captures the minimum coherence between the current gradient $\nabla F(\mathbf{x}_k)$ and the gradients along the past $s$ iterations[8]. Intuitively, if $\mu_k$ is positive, then the direction of the current gradient is well aligned to those of the past gradients. In this case, the convergence property induced by using delayed stochastic gradients is close to that induced by using synchronous stochastic gradients. Note that Definition 1 only requires the gradients to be positively correlated over a small number of iterations s, which is often very small (e.g. <10 in our experiments). Therefore, Definition 1 is *not* a global requirement on optimization path.

Even though neural network's loss function is non-convex, recent studies showed strong evidences that SGD in practical neural network training encourage positive gradient coherence (Li et al., 2017; Lorch, 2016). This is consistent with the findings that the loss surface of shallow networks and deep networks with skip connections are dominated by large, flat, nearly convex attractors around the critical points (Li et al., 2017; Keskar et al., 2016), implying that the degree of non-convexity is mild around critical points. We show in the sequel that $\mu_k > 0$ through most of the optimization path, especially when the staleness is minimized in practice by system optimization (Fig. 4). Our theory can be readily adapted to account for a limited amount of negative $\mu_k$ (see Appendix), but our primary interest is to provide a quantity that is (1) easy to compute empirically during the course of optimization[9], and (2) informative for the impact of staleness and can potentially be used to control synchronization levels. We now characterize the convergence property of Async-SGD.

**Theorem 1.** *Let Assumption 1 hold. Suppose for some $\mu > 0$, the gradient coherence satisfies $\mu_k \geq \mu$ for all $k$ and the variance of the stochastic gradients is bounded by $\sigma^2 > 0$. Choose stepsize $\eta_k = \frac{\mu}{sL\sqrt{k}}$. Then, the iterates generated by the Async-SGD satisfy*

$$\min_{0 \leq k \leq T} \mathbb{E}\|\nabla F(\mathbf{x}_k)\|^2 \leq \left( \frac{sL(F(\mathbf{x}_0) - \inf_{\mathbf{x}} F(\mathbf{x}))}{\mu^2} + \frac{\sigma^2 \log T}{s} \right) \frac{1}{\sqrt{T}}. \tag{1}$$

---

[8]Our gradient coherence bears similarity with the sufficient direction assumption in (Huo et al., 2018). However, sufficient direction is a layer-wise and fixed delay, whereas our staleness is a random variable that is subject to system level factors such as communication bandwidth

[9]It can be approximated by storing a pre-selected batch of data on a worker. The worker just needs to compute gradient every $T$ mini-batches to obtain approximate $\nabla F(\mathbf{x}_k), \nabla F(\mathbf{x}_t)$ in Definition 1.

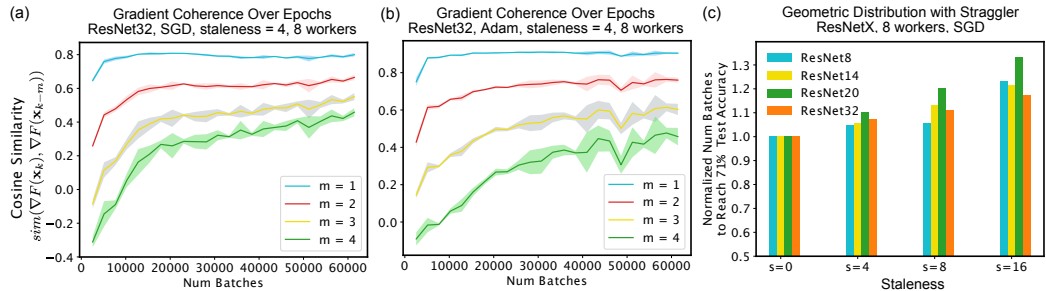

Figure 4: (a)(b) Cosine similarity between the gradient at the $k$-th iteration $\nabla F(\mathbf{x}_k)$, and the gradient $m$ steps prior $\nabla F(\mathbf{x}_{k-m})$, over the course of convergence for ResNet32 on CIFAR10 optimized by SGD (a) and Adam (b) under staleness $s = 4$ on 8 workers with parameters in Table 1. Shaded region is 1 standard deviation over 3 runs. For computational efficiency, we approximate the full gradient $\nabla F(\mathbf{x}_k)$ by gradients on a fixed set of 1000 training samples $D_{fixed}$ and use $\nabla_{D_{fixed}} F(\mathbf{x}_k)$. (c) The number of batches to reach 71% test accuracy on CIFAR10 for ResNet8-32 using 8 workers and SGD under geometric delay distribution (details in Appendix).

We refer readers to Appendix for the the proof. Theorem 1 characterizes several theoretical aspects of Async-SGD. First, the choice of the stepsize $\eta_k = \frac{\mu}{sL\sqrt{k}}$ is adapted to both the maximum staleness and the gradient coherence. Intuitively, if the system encounters a larger staleness, then a smaller stepsize should be used to compensate the negative effect. On the other hand, the stepsize can be accordingly enlarged if the gradient coherence along the iterates turns out to be high. In this case, the direction of the gradient barely changes along the past several iterations, and a more aggressive stepsize can be adopted. In summary, the choice of stepsize should trade-off between the effects caused by both the staleness and the gradient coherence.

Furthermore, Theorem 1 shows that the minimum gradient norm decays at the rate $\mathcal{O}(\frac{\log T}{\sqrt{T}})$, implying that the Async-SGD converges to a stationary point provided a positive gradient coherence, which we observe empirically in the sequel. On the other hand, the bound in Eq. (1) captures the trade-off between the maximum staleness $s$ and the gradient coherence $\mu$. Specif-

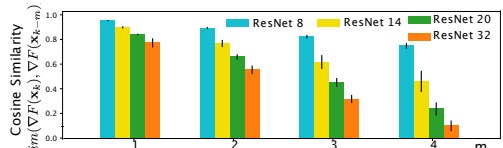

Figure 5: Gradient coherence for ResNet with varying depths optimized by SGD using 8 workers. The x-axis $m$ is defined in Fig. 4

ically, minimizing the right hand side of Eq. (1) with regard to the maximum staleness $s$ yields the optimal choice $s^* = \sigma\mu\sqrt{\frac{\log T}{L(F(\mathbf{x}_0) - \inf_{\mathbf{x}} F(\mathbf{x}))}}$, i.e., a larger staleness is allowed if the gradients remain to be highly coherent along the past iterates.

**Empirical Observations.** Theorem 1 suggests that more coherent gradients along the optimization paths can be advantageous under non-synchronous execution. Fig. 4 shows the cosine similarity $sim(\mathbf{a}, \mathbf{b}) := \frac{\mathbf{a} \cdot \mathbf{b}}{\|\mathbf{a}\|\|\mathbf{b}\|}$ between gradients along the convergence path for CNNs and DNNs[10]. We observe the followings: (1) Cosine similarity improves over the course of convergence (Fig. 4(a)(b)). Except the highest staleness during the early phase of convergence, cosine similarity remains positive[11]. In practice the staleness experienced during run time can be limited to small staleness (Dai et al., 2015), which minimizes the likelihood of negative gradient coherence during the early phase. (2) Fig. 5 shows that cosine similarity decreases with increasing CNN model complexity. Theorem 1 implies that lower gradient coherence amplifies the effect of staleness $s$ through the factor $\frac{s}{\mu^2}$ in Eq. (1). This is consistent with the convergence difficulty encountered in deeper models (Fig. 1).

## 6 DISCUSSION AND CONCLUSION

In this work, we study the convergence behaviors under delayed updates for a wide array of models and algorithms. Our extensive experiments reveal that staleness appears to be a key governing parameter in learning. Overall staleness slows down the convergence, and under high staleness levels the convergence can progress very slowly or fail. The effects of staleness are highly problem

---

[10]Cosine similarity is closely related to the coherence measure in Definition 1.

[11]Low gradient coherence during the early part of optimization is consistent with the common heuristics to use fewer workers at the beginning in asynchronous training. (Lian et al., 2015) also requires the number of workers to follow $\frac{1}{\sqrt{K}}$ where $K$ is the iteration number.

dependent, influenced by model complexity, choice of the algorithms, the number of workers, and the model itself, among others. Our empirical findings inspire new analyses of non-convex optimization under asynchrony based on gradient coherence, matching the existing rate of $\mathcal{O}(1/\sqrt{T})$.

Our findings have clear implications for distributed ML. To achieve actual speed-up in absolute convergence, any distributed ML system needs to overcome the slowdown from staleness, and carefully trade off between system throughput gains and statistical penalties. Many ML methods indeed demonstrate certain robustness against low staleness, which should offer opportunities for system optimization. Our results support the broader observation that existing successful non-synchronous systems generally keep staleness low and use algorithms efficient under staleness.

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

# A APPENDIX

## A.1 PROOF OF THEOREM 1

**Theorem 2.** *Let Assumption 1 hold. Suppose the gradient coherence $\mu_k$ is lower bounded by some $\mu > 0$ for all $k$ and the variance of the stochastic gradients is upper bounded by some $\sigma^2 > 0$. Choose stepsize $\eta_k = \frac{\mu}{sL\sqrt{k}}$. Then, the iterates generated by the Async-SGD satisfy*

$$\min_{0 \le k \le T} \mathbb{E}\|\nabla F(\mathbf{x}_k)\|^2 \le \Big( \frac{sL(F(\mathbf{x}_0) - \inf_{\mathbf{x}} F(\mathbf{x}))}{\mu^2} + \frac{\sigma^2 \log T}{s} \Big) \frac{1}{\sqrt{T}}. \tag{2}$$

*Proof.* By the $L$-Lipschitz property of $\nabla F$, we obtain that for all $k$

$$F(\mathbf{x}_{k+1}) \le F(\mathbf{x}_k) + \langle \mathbf{x}_{k+1} - \mathbf{x}_k, \nabla F(\mathbf{x}_k) \rangle + \frac{L}{2} \|\mathbf{x}_{k+1} - \mathbf{x}_k\|^2 \tag{3}$$

$$= F(\mathbf{x}_k) - \eta_k \langle \nabla f_{\xi(\tau_k)}(\mathbf{x}_{\tau_k}), \nabla F(\mathbf{x}_k) \rangle + \frac{L\eta_k^2}{2} \|\nabla f_{\xi(\tau_k)}(\mathbf{x}_{\tau_k})\|^2. \tag{4}$$

Taking expectation on both sides of the above inequality and note that the variance of the stochastic gradient is bounded by $\sigma^2$, we further obtain that

$$\mathbb{E}[F(\mathbf{x}_{k+1})] \le \mathbb{E}[F(\mathbf{x}_k)] - \eta_k \mathbb{E}[\langle \nabla F(\mathbf{x}_{\tau_k}), \nabla F(\mathbf{x}_k) \rangle] + \frac{L\eta_k^2}{2} \mathbb{E}[\|\nabla F(\mathbf{x}_{\tau_k})\|^2 + \sigma^2] \tag{5}$$

$$\le \mathbb{E}[F(\mathbf{x}_k)] - \eta_k \mu_k \mathbb{E}\|\nabla F(\mathbf{x}_k)\|^2 + \frac{L\eta_k^2}{2} \mathbb{E}\|\nabla F(\mathbf{x}_{\tau_k})\|^2 + \frac{\sigma^2 L\eta_k^2}{2} \tag{6}$$

$$\le \mathbb{E}[F(\mathbf{x}_k)] - \eta_k \mu \mathbb{E}\|\nabla F(\mathbf{x}_k)\|^2 + \frac{L\eta_k^2}{2} \sum_{t=k-s+1}^{k} \mathbb{E}\|\nabla F(\mathbf{x}_t)\|^2 + \frac{\sigma^2 L\eta_k^2}{2}. \tag{7}$$

Telescoping the above inequality over $k$ from $0$ to $T$ yields that

$$\mathbb{E}[F(\mathbf{x}_{k+1})] - \mathbb{E}[F(\mathbf{x}_0)]$$

$$\le - \sum_{k=0}^{T} \eta_k \mu \mathbb{E}\|\nabla F(\mathbf{x}_k)\|^2 + \frac{L}{2} \sum_{k=0}^{T} \sum_{t=k-s+1}^{k} \eta_k^2 \mathbb{E}\|\nabla F(\mathbf{x}_t)\|^2 + \frac{\sigma^2 L}{2} \sum_{k=0}^{T} \eta_k^2 \tag{8}$$

$$\le - \sum_{k=0}^{T} \eta_k \mu \mathbb{E}\|\nabla F(\mathbf{x}_k)\|^2 + \frac{L}{2} \sum_{k=0}^{T} \sum_{t=k-s+1}^{k} \eta_t^2 \mathbb{E}\|\nabla F(\mathbf{x}_t)\|^2 + \frac{\sigma^2 L}{2} \sum_{k=0}^{T} \eta_k^2 \tag{9}$$

$$\le - \sum_{k=0}^{T} \eta_k \mu \mathbb{E}\|\nabla F(\mathbf{x}_k)\|^2 + \frac{Ls}{2} \sum_{k=0}^{T} \eta_k^2 \mathbb{E}\|\nabla F(\mathbf{x}_k)\|^2 + \frac{\sigma^2 L}{2} \sum_{k=0}^{T} \eta_k^2 \tag{10}$$

$$= \sum_{k=0}^{T} \Big( \frac{Ls\eta_k^2}{2} - \eta_k \mu \Big) \mathbb{E}\|\nabla F(\mathbf{x}_k)\|^2 + \frac{\sigma^2 L}{2} \sum_{k=0}^{T} \eta_k^2. \tag{11}$$

Rearranging the above inequality and note that $F(\mathbf{x}_{k+1}) > \inf_{\mathbf{x}} F(\mathbf{x}) > -\infty$, we further obtain that

$$\sum_{k=0}^{T} \Big( \eta_k \mu - \frac{Ls\eta_k^2}{2} \Big) \mathbb{E}\|\nabla F(\mathbf{x}_k)\|^2 \le \big( F(\mathbf{x}_0) - \inf_{\mathbf{x}} F(\mathbf{x}) \big) + \frac{\sigma^2 L}{2} \sum_{k=0}^{T} \eta_k^2. \tag{12}$$

Note that the choice of stepsize guarantees that $\eta_k \mu - \frac{Ls\eta_k^2}{2} > 0$ for all $k$. Thus, we conclude that

$$\min_{0 \le k \le T} \mathbb{E}\|\nabla F(\mathbf{x}_k)\|^2 \le \frac{\big( F(\mathbf{x}_0) - \inf_{\mathbf{x}} F(\mathbf{x}) \big) + \frac{\sigma^2 L}{2} \sum_{k=0}^{T} \eta_k^2}{\sum_{k=0}^{T} \big( \eta_k \mu - \frac{Ls\eta_k^2}{2} \big)} \tag{13}$$

$$\le \frac{2\big( F(\mathbf{x}_0) - \inf_{\mathbf{x}} F(\mathbf{x}) \big) + \sigma^2 L \sum_{k=0}^{T} \eta_k^2}{\sum_{k=0}^{T} \eta_k \mu}, \tag{14}$$

where the last inequality uses the fact that $\eta_k \mu - \frac{L s \eta_k^2}{2} > \frac{\eta_k \mu}{2}$. Substituting the stepsize $\eta_k = \frac{\mu}{sL\sqrt{k}}$ into the above inequality and simplifying, we finally obtain that

$$\min_{0 \leq k \leq T} \mathbb{E}\|\nabla F(\mathbf{x}_k)\|^2 \leq \left( \frac{sL\big(F(\mathbf{x}_0) - \inf_{\mathbf{x}} F(\mathbf{x})\big)}{\mu^2} + \frac{\sigma^2 \log T}{s} \right) \frac{1}{\sqrt{T}}. \quad (15)$$

$\square$

## A.2 Handling Negative Gradient Coherence in Theorem 1

Our assumption of positive gradient coherence (GC) is motivated by strong empirical evidence that GC is largely positive (Fig. 4(a)(b) in the main text). Contrary to conventional wisdom, GC generally *improves* when approaching convergence for both SGD and Adam. Furthermore, in practice, the effective staleness for any given iteration generally concentrates in low staleness for the non-stragglers (Dai et al., 2015).

When some $\mu_k$ are negative at some iterations, in eq. 11 in the Appendix we can move the negative terms in $\sum_k \eta_k \mu_k$ to the right hand side and yield a higher upper bound (i.e., slower convergence). This is also consistent with empirical observations that higher staleness lowers GC and slows convergence.

## A.3 Exponential delay distribution.

We consider delays drawn from geometric distribution (GD), which is the discrete version of exponential distribution. For each iterate we randomly select a worker to be the straggler with large mean delay ($p = 0.1$), while all other non-straggler workers have small delays. The non-straggler delay is drawn from GD with $p$ chosen to achieve the same mean delay as in the uniform case (after factoring in straggler) in the main text. The delay is drawn per worker for each iteration, and thus a straggler's outgoing updates to all workers suffer the same delay. Fig. 4(c) in the main text shows the convergence speed under the corresponding staleness $s$ with the same mean delay (though $s$ is not a parameter in GD). It exhibits trends analogous to Fig. 1(b) in the main text: staleness slows convergence substantially and overall impacts deeper networks more.

## A.4 Additional Results for DNNs

We present additional results for DNNs. Fig. 6 shows the number of batches, normalized by $s = 0$, to reach convergence using 1 hidden layer and 1 worker under varying staleness levels and batch sizes. Overall, the effect of batch size is relatively small except in high staleness regime ($s = 32$).

Fig. 7 shows the number of batches to reach convergence, normalized by $s = 0$ case, for 5 variants of SGD using 1 worker. The results are in line with the analyses in the main text: staleness generally leads to larger slow down for deeper networks than shallower ones. SGD and Adagrad are more robust to staleness than Adam, RMSProp, and SGD with momentum. In particular, RMSProp exhibit high variance in batches to convergence (not shown in the normalized plot) and thus does not exhibit consistent trend.

Fig. 8 shows the number of batches to convergence under Adam and SGD on 1, 8, 16 simulated workers, respectively normalized by staleness 0's values. The results are consistent with the observations and analyses in the main text, namely, that having more workers amplifies the effect of staleness. We can also observe that SGDS is more robust to staleness than Adam, and shallower networks are less impacted by staleness. In particular, note that staleness sometimes accelerates convergence, such as in Fig. 8(d). This is due to the implicit momentum created by staleness (Mitliagkas et al., 2016).

## A.5 LDA and Additional Results for LDA

In LDA each token $w_{ij}$ ($j$-th token in the $i$-th document) is assigned with a latent topic $z_{ij}$ from totally $K$ topics. We use Gibbs sampling to infer the topic assignments $z_{ij}$. The Gibbs sampling step involves three sets of parameters, known as sufficient statistics: (1) document-topic vector $\theta_i \in \mathbb{R}^K$ where $\theta_{ik}$ the number of topic assignments within document $i$ to topic $k = 1...K$; (2) word-topic

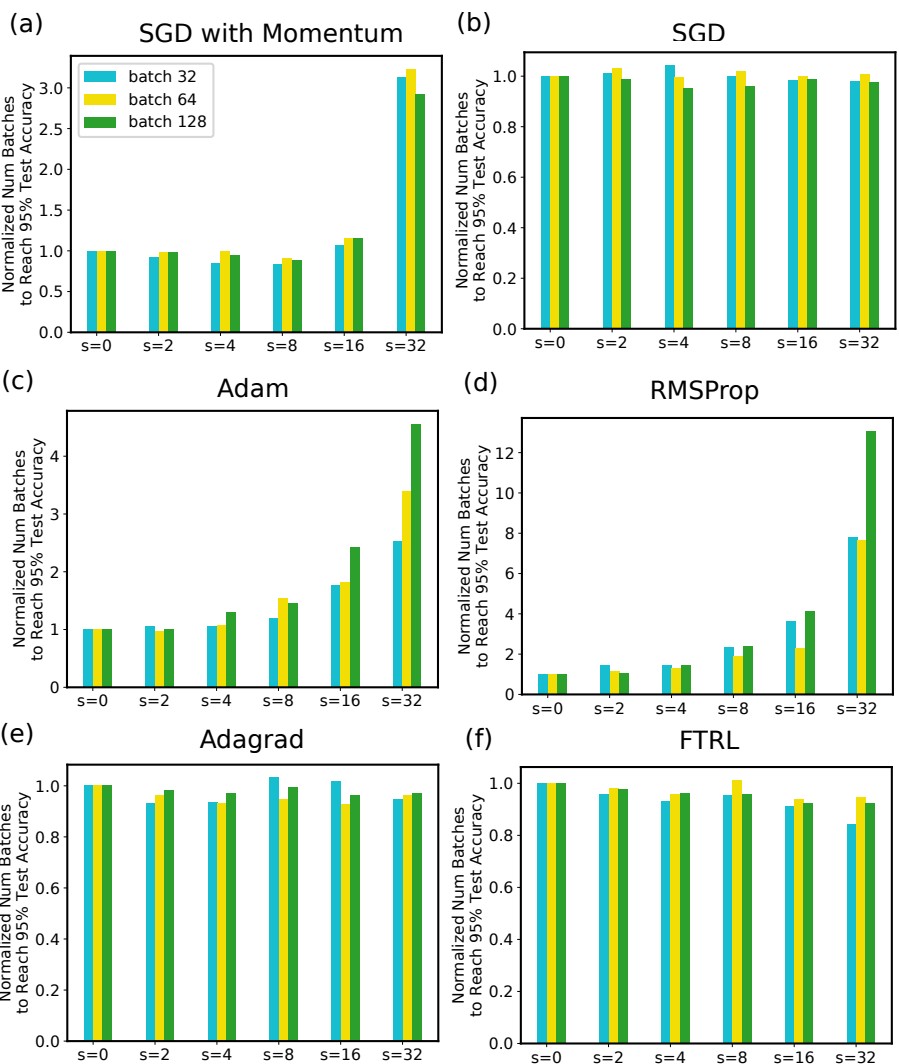

Figure 6: The number of batches to reach 95% test accuracy using 1 hidden layer and 1 worker, respectively normalized by $s = 0$.

vector $\phi_w \in \mathbb{R}^K$ where $\phi_{wk}$ is the number of topic assignments to topic $k = 1, ..., K$ for word (vocabulary) $w$ across all documents; (3) $\tilde{\phi} \in \mathbb{R}^K$ where $\tilde{\phi}_k = \sum_{w=1}^W \phi_{wk}$ is the number of tokens in the corpus assigned to topic $k$. The corpus $(w_{ij}, z_{ij})$ is partitioned to workers, while $\phi_w$ and $\tilde{\phi}$ are shared model parameters. We measure the model quality using log likelihood.

We present additional results of LDA under different numbers of workers and topics in Fig. 9 and Fig. 10. These panels extends Fig. 3(c)(d) in the main text. See the main text for experimental setup and analyses and experimental setup.

## A.6 Additional Results for MF

We show the convergence curves for MF under different numbers of workers and staleness levels in Fig. 11. It is evident that higher staleness leads to a higher variance in convergence. Furthermore, the number of workers also affects variance, given the same staleness level. For example, MF with 4 workers incurs very low standard deviation up to staleness 20. In contrast, MF with 8 workers already exhibits a large variance at staleness 15. The amplification of staleness from increasing number of

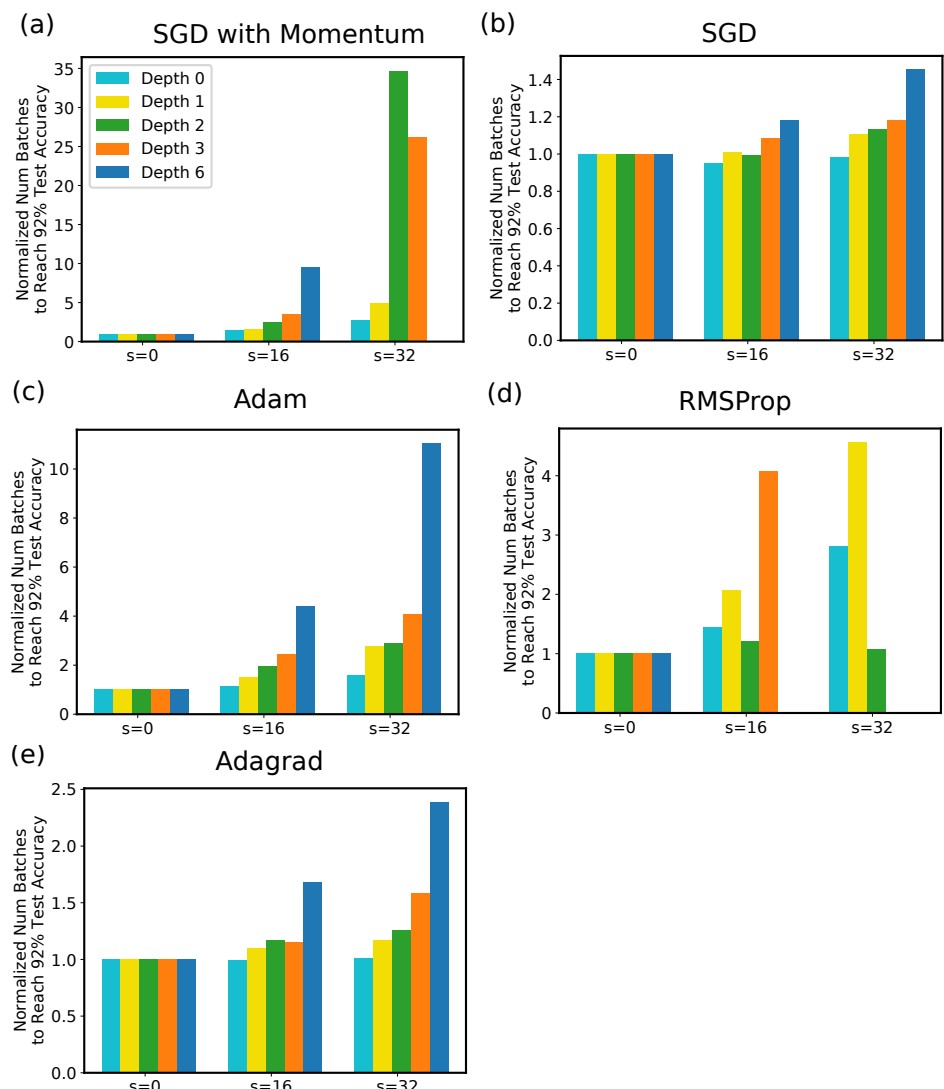

Figure 7: The number of batches to reach 92% test accuracy using DNNs with varying numbers of hidden layers under 1 worker. We consider several variants of SGD algorithms (a)-(e). Note that with depth 0 the model reduces to MLR, which is convex. The numbers are averaged over 5 randomized runs. We omit the result whenever convergence is not achieved within the experiment horizon (77824 batches), such as SGD with momentum at depth 6 and $s = 32$.

workers is consistent with the discussion in the main text. See the main text for experimental setup and analyses.

### A.7 ADDITIONAL RESULTS FOR VAEs

Fig. 12 shows the number of batches to reach test loss 130 by Variational Autoencoders (VAEs) on 1 worker, under staleness 0 to 16 and 4 SGD variants. We consider VAEs with depth 1, 2, and 3 (the number of layers in the encoder and decoder networks). The number of batches are normalized by $s = 0$ for each VAE depth, respectively. See the main text for analyses.

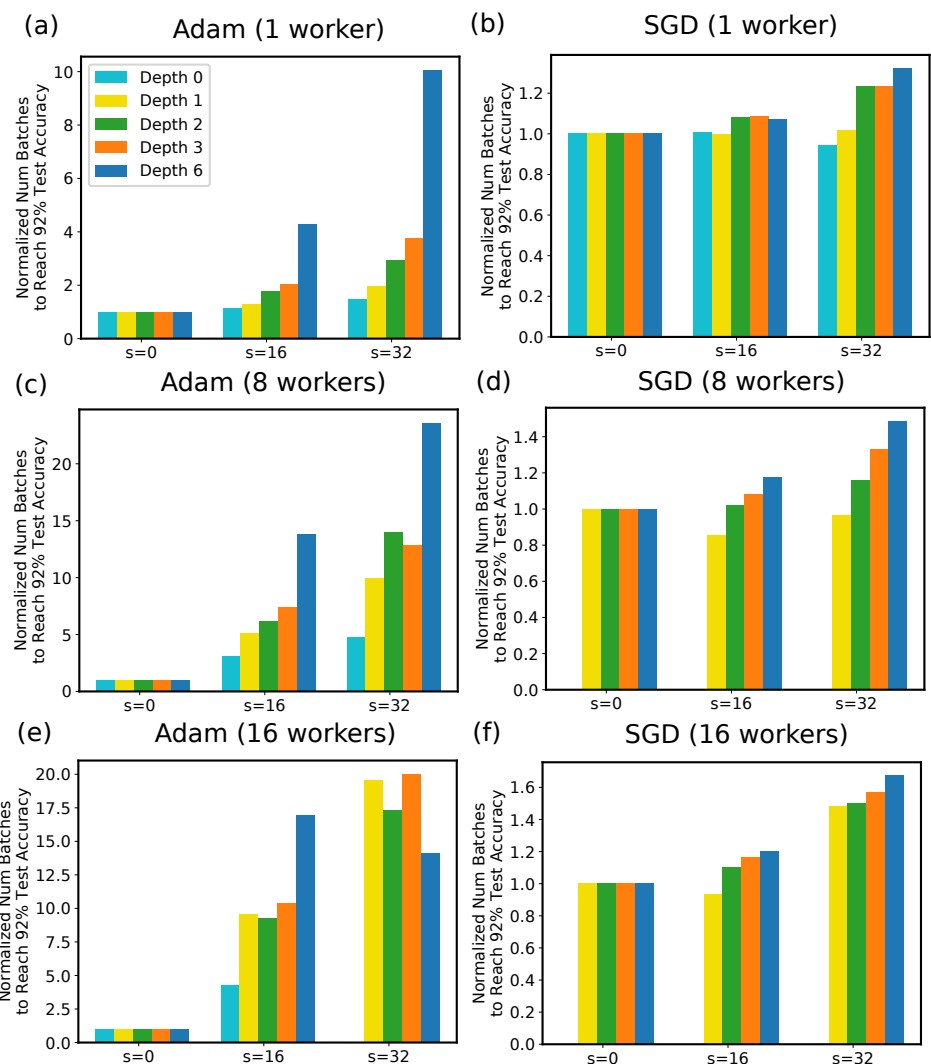

Figure 8: The number of batches to reach 92% test accuracy with Adam and SGD on 1, 8, 16 workers with varying staleness. Each model depth is normalized by the staleness 0's values, respectively. The numbers are average over 5 randomized runs. Depth 0 under SGD with 8 and 16 workers did not converge to target test accuracy within the experiment horizon (77824 batches) for all staleness values, and is thus not shown.

## A.8 RECURRENT NEURAL NETWORKS

Recurrent Neural Networks (RNNs) are widely used in recent natural language processing tasks. We consider long short-term memory (LSTM) (Hochreiter & Schmidhuber, 1997) applied to the language modeling task, using a subset of Penn Treebank dataset (PTB) (Marcus et al., 1993) containing 5855 words. The dataset is pre-processed by standard de-capitalization and tokenization. We evaluate the impact of staleness for LSTM with 1 to 4 layers, with 256 neurons in each layer. The maximum length for each sentence is 25. Note that 4 layer LSTM is about 4x more model parameters than the 1 layer LSTM, which is the same ratio between ResNet32 and Resnet 8. We use batch size 32 similar to other experiments. We consider staleness $s = 0, 4, 8, 16$ on 8 workers. The model quality is measured in perplexity. Fig. 13 shows the number of batches needed to reach the desired model quality for RNNs with varying network depths. We again observe that staleness impacts deeper network variants more than shallower counterparts, which is consistent with our observation in CNNs and DNNs.

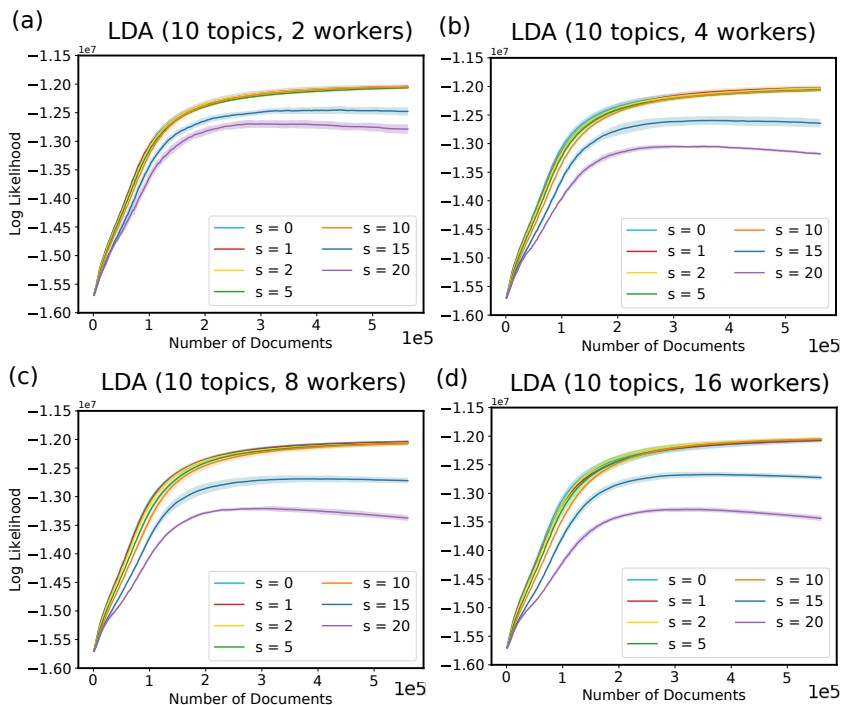

Figure 9: Convergence of LDA log likelihood using 10 topics with respect to the number of documents processed by collapsed Gibbs sampling, with varying staleness levels and number of workers. The shaded regions are 1 standard deviation around the means (solid lines) based on 5 randomized runs.

Fig. 14 shows the number of batches needed to reach the desired model quality for RNNs with on 4 SGD variants: SGD, Adam, Momentum, and RMSProp. Similar to the discussion in the main text, different algorithms respond to staleness differently, with SGD and Adam more robust to staleness than Momentum and RMSProp.[12]

---

[12]We however, note that we have not tuned the learning rate in this experiment. RMSProp might benefit from a lower learning rate at high staleness.

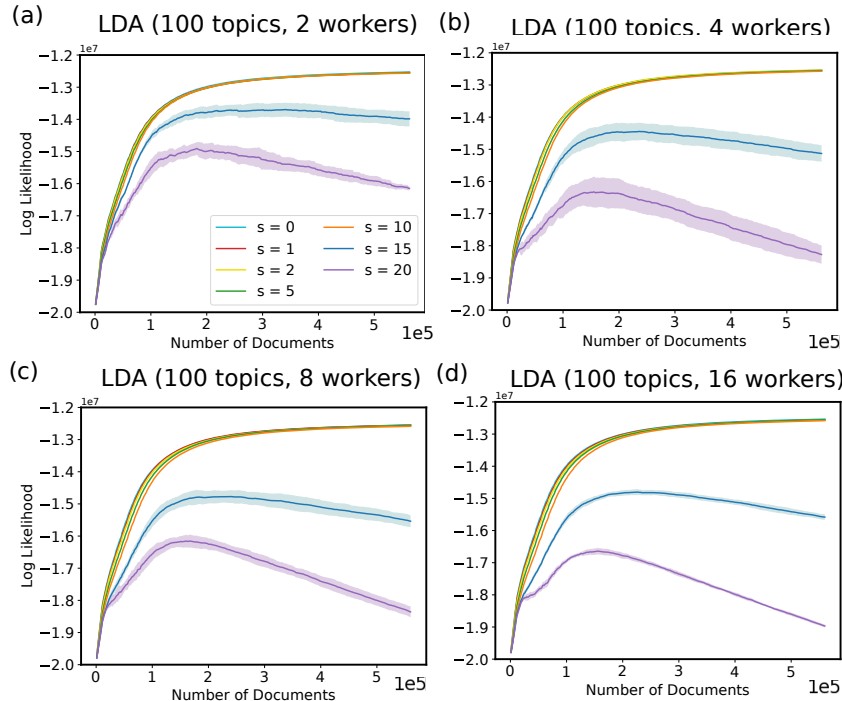

Figure 10: Convergence of LDA log likelihood using 100 topics with respect to the number of documents processed by collapsed Gibbs sampling, with varying staleness levels and the number of workers. The shaded regions are 1 standard deviation around the means (solid lines) based on 5 randomized runs.

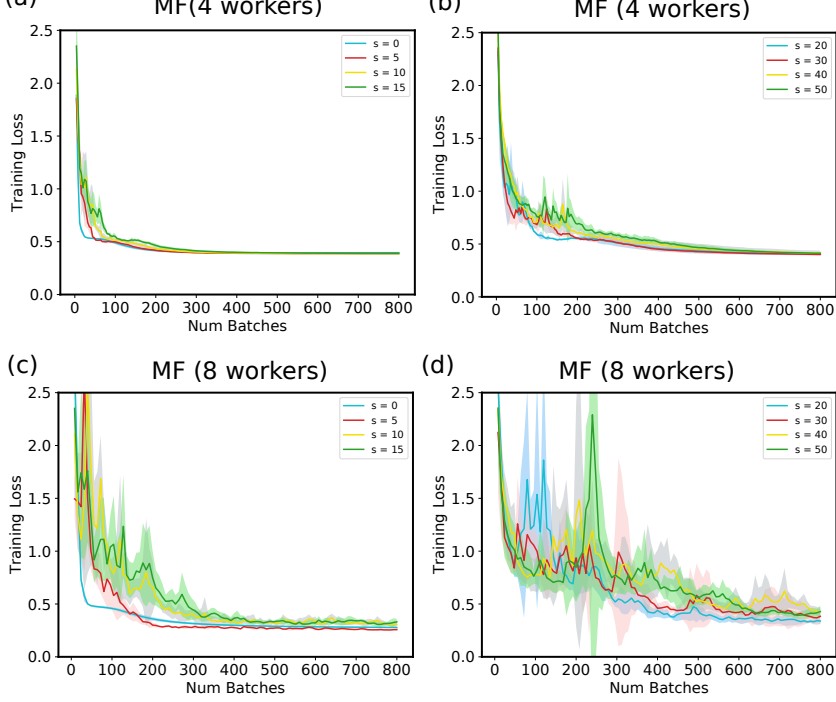

Figure 11: Convergence of Matrix Factorization (MF) using 4 and 8 workers, with staleness ranging from 0 to 50. The x-axis shows the number of batches processed across all workers. Shaded area represents 1 standard deviation around the means (solid curves) computed on 5 randomized runs.

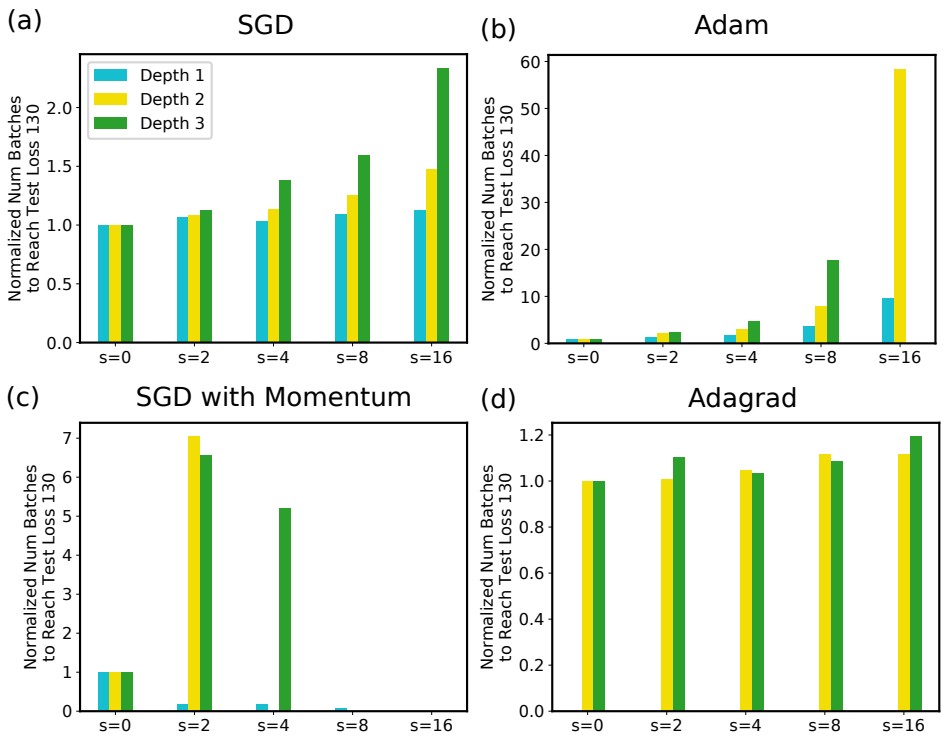

Figure 12: The number of batches to reach test loss 130 by Variational Autoencoders (VAEs) on 1 worker, under staleness 0 to 16. We consider VAEs with depth 1, 2, and 3 (the number of layers in the encoder and the decoder networks). The numbers of batches are normalized by $s = 0$ for each VAE depth, respectively. Configurations that do not converge to the desired test loss are omitted, such as Adam optimization for VAE with depth 3 and $s = 16$.

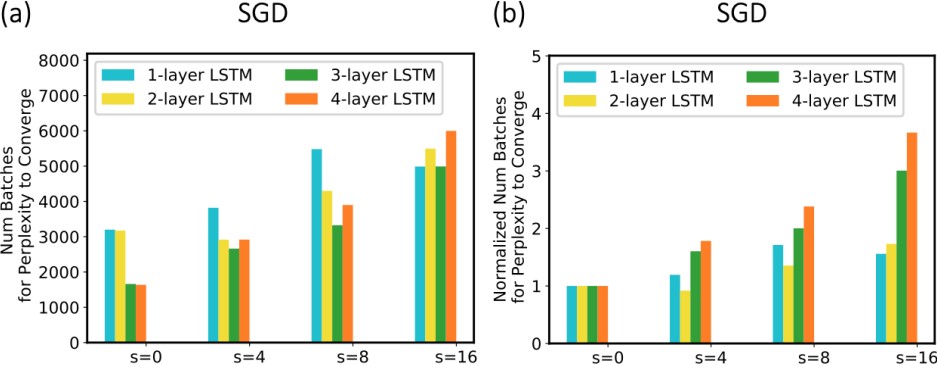

Figure 13: The number of batches to converge (measured using perplexity) for RNNs on 8 workers, under staleness 0 to 16. We consider LSTMs with depth $\{1, 2, 3, 4\}$. See section A.8 for details.

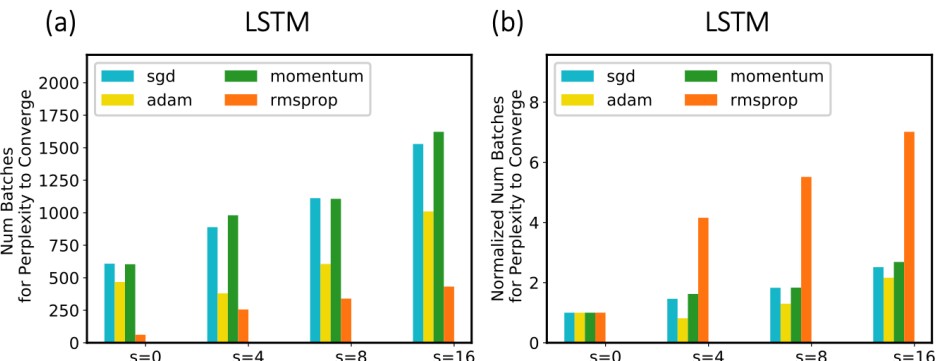

Figure 14: The number of batches to converge ((measured using perplexity)) for RNNs on 8 workers, under staleness 0 to 16. We consider 4 variants of SGD: vanilla SGD, Adam, Momentum, RMSProp. See section A.8 for details.

