# OpenReview forum: "Toward Understanding the Impact of Staleness in Distributed Machine Learning"
_ICLR.cc/2019/Conference_

### Official Review · AnonReviewer3 · 2018-10-16
**Empirical explanation of the impact of staleness**

**Rating:** 7
**Confidence:** 5

**Review:**

This paper tries to analyze the impact of the staleness on machine learning models in different settings, including model complexity, optimization methods or the number of workers. In this work, they study the convergence behaviors of a wide array of ML models and algorithms under delayed updates, and propose a new convergence analysis of asynchronous SGD method for non-convex optimization.

The following are my concerns:
1. "For CNNs and DNNs, the staleness slows down deeper models much more than shallower counterparts." I think it is straightforward. I want to see the theoretical analysis of the relation between model complexity and staleness.
2. "Different algorithms respond to staleness very differently".  This finding is quite interesting. Is there any theoretical analysis of this phenomenon?
3. The "gradient coherence"  in the paper is not new. I am certain that "gradient coherence" is very similar to the "sufficient direction" in [1].
4. What is the architecture of the network? in the paper, each worker p can communicate with other workers p'. Does it mean that it is a grid network? or it is just a start network.
5. in the top of page 3, why the average delay under the model is 1/2s +1, isn't it (s-1)/2?
6.  on page 5, "This is perhaps not surprising, given the fact that deeper models pose more optimization challenges even under the sequential settings." why it is obvious opposite to your experimental results in figure 1(a)? Could you explain why shallower CNN requires more iterations to get the same accuracy? it is a little counter-intuitive.
7. I don't understand what does "note that s = 0 execution treats each worker’s update as separate updates instead of one large batch in other synchronous systems" mean in the footnote of page 5.


Above all, this paper empirically analyzes the effect of the staleness on the model and optimization methods. It would be better if there is some theoretical analysis to support these findings.

[1] Training Neural Networks Using Features Replay  https://arxiv.org/pdf/1807.04511.pdf


===after rebuttal===
All my concerns are addressed. I will upgrade the score.

---

> ### Author Response · Authors · 2018-11-19
> **Response to AnonReviewer3**
>
> We thank the reviewer for the valuable feedback. Our work aims to strike a balance between empirical and theoretical approaches to understanding the effects of stale updates. Our goals in this work is threefold: (1) Through systematic experiments, we explicitly observe staleness and its impact, for the first time to our knowledge, on 12 key models and algorithms. (2) We introduce gradient coherence (GC), which is related to the impact of staleness for gradient-based optimization. GC can be evaluated in real time during the course of convergence, with minimal overhead, and may be used by practitioners to control delays in the system. (3) Based on GC, we provide a new convergence analysis of SGD in non-convex optimization under staleness. With such a broad scope, there is inevitably areas for improvements. We hope that the reviewer will consider the contributions towards making distributed ML more robust under non-synchronous execution as we address the comments:
>
> 1. The reviewer indeed raised interesting points. Our theory based on gradient coherence relates model complexity to the larger slowdown by staleness through the gradient coherence. Fig. 5 in the manuscript shows that deeper network generally exhibits lower gradient coherence. Our theorem shows that lower gradient coherence amplifies the effect of staleness s through the factor s/mu^2 in Eq (1) in the manuscript. We have included a brief discussion of this point in the manuscript.
>
> 2. Staleness is known to add implicit momentum to SGD gradients [2]. The Adam optimizer keeps an exponentially decaying average of past gradients to modify gradient direction, and can be viewed as a version of momentum methods, whose momentum may be affected by staleness by similar reasoning. It is, however, challenging to analyze the convergence of these advanced gradient descent methods even under sequential settings [3], and the treatment under staleness is beyond the scope of our current work. It’d be an interesting future direction to create a delay tolerant version of Adam, similar to AdaRevision [4].
>
> 3. We thank the reviewer for pointing out a reference that we were not aware of. We agree that the sufficient direction assumption in [1] shares resemblance to our Definition 1. We note that their ``staleness’’ in the definition of sufficient direction is based on a layer-wise and fixed delay, whereas our staleness is a random variable that is subject to system level factors such as communication bandwidth. Also, we note that their convergence results in Theorem 1 and Theorem 2 do not capture the impact of staleness, whereas our Theorem 1 explicitly characterizes its impact on the choice of stepsize and the convergence rate, and also captures the interplay to gradient coherence. We have included the reference in our updated manuscript to provide further context.
>
> 4. Though we use a peer to peer topology in our experiment, our delay pattern is agnostic to the underlying communication network topology. In practice it is more common to implement an intermediate aggregation such as parameter server [5] to reduce network traffic.
>
> 5. We thank the reviewer for pointing out the error. The delay should be r ~ Categorical(0, 1, …, s), which gives the 0.5s + 1 expected delay. We have corrected in the updated manuscript.
>
> 6. This is an important point to clarify. With SGD, ResNet8’s final test accuracy is about 73% in our setting, while ResNet20’s final test accuracy is close to 75%. Therefore, deeper ResNet can reach the same model accuracy in the earlier part of the optimization path, resulting in lower number of batches in Fig.1(a). However, when the convergence time is normalized by the non-stale (s=0) value, we observe the impact of staleness is higher on deeper models. We have included this clarification in the updated manuscript.
>
> 7. Many synchronous systems uses batch size linear in the number of workers (e.g., [6]). We preserve the same batch size and more workers simply makes more updates in each iteration. We have reworded the footnote for better clarity.
>
> [1] Training Neural Networks Using Features Replay.  https://arxiv.org/pdf/1807.04511.pdf
> [2] Ioannis Mitliagkas et al. Asynchrony begets momentum, with an application to deep learning.
> [3] Sashank J. Reddi, Satyen Kale, and Sanjiv Kumar. On the convergence of adam and beyond. International Conference on Learning Representations, 2018.
> [4] H. Brendan Mcmahan and Matthew Streeter. Delay-Tolerant Algorithms for Asynchronous Distributed Online Learning. NIPS 2014.
> [5] M. Li, D. G. Andersen, J. Park, A. J. Smola, A. Ahmed, V. Josifovski, J. Long, E. J. Shekita, and B.-Y. Su. Scaling distributed machine learning with the Parameter Server. In Proceedings of OSDI, 2014.
> [6] P. Goyal and et al. ´ A. Kyrola, A. Tulloch, Y. Jia, and K. He, “Accurate, large minibatch SGD: training imagenet in 1 hour,” CoRR, vol. abs/1706.02677, 2017.

---

### Official Review · AnonReviewer1 · 2018-10-25
**Interesting empirical and theoretical analysis of the convergence of async SGD under delay**

**Rating:** 9
**Confidence:** 4

**Review:**

This paper presents and empirical and theoretical study of the convergence of asynchronous stochastic gradient descent training if there are delays due to the asynchronous part of it. The paper can be neatly split in two parts: a simulation study and a theoretical analysis.

The simulation study compares, under fixed hyperparameters, the behavior of distributed training under different simulated levels of delay on different problems and different model architectures. Overall the results are very interesting, but the simulation could have been more thorough. Specifically, the same hyperparameter values were used across batch sizes and across different values of the distributed delay. Some algorithms failed to converge under some settings and others experienced dramatic slowdowns, but without careful study of hyperparameters it's hard to tell whether these behaviors are normal or outliers. Also it would have been interesting to see a recurrent architecture there, as I've heard much anecdotal evidence about the robustness of RNNs and LSTMs to asynchronous training. I strongly advise the authors to redo the experiments with some hyperparameter tuning for different levels of staleness to make these results more believable.

The theoretical analysis identifies a quantity called gradient coherence and proves that a learning rate based on the coherence can lead to an optimal convergence rate even under asynchronous training. The proof is correct (I checked the major steps but not all details), and it's sufficiently different from the analysis of hogwild algorithms to be of independent interest. The paper also shows the empirical behavior of the gradient coherence statistic during model training; interestingly this seems to also explain the heuristic commonly believed that to make asynchronous training work one needs to slowly anneal the number of workers (coherence is much worse in the earlier than later phases of training). This quantity is interesting also because it's somewhat independent of the variance of the stochastic gradient across minibatches (it's the time variance, in a way), and further analysis might also show interesting results.

---

> ### Author Response · Authors · 2018-11-19
> **Response to AnonReviewer1**
>
> We appreciate the insightful comments and careful review of our work. Our goals in this work is threefold: (1) Through systematic experiments, we explicitly observe staleness and its impact, for the first time to our knowledge, on 12 key models and algorithms. (2) We introduce gradient coherence (GC), which is related to the impact of staleness for gradient-based optimization. GC can be evaluated in real time during the course of convergence, with minimal overhead, and may be used by practitioners to control delays in the system. (3) Based on GC, we provide a new convergence analysis of SGD in non-convex optimization under staleness. With such a broad scope, there is inevitably areas for improvements. We hope that the reviewer will consider the contributions towards making distributed ML more robust under non-synchronous execution as we address the comments:
>
> Regarding to fixed hyperparameters: we have redone all experiments in Fig. 2 with hyperparameter search over the learning rate. We observe the same overall pattern as before: staleness slows down convergence, sometimes quite significantly at high levels of staleness. Furthermore, different algorithms have different sensitivity to staleness, and show similar trends as observed before. For example, SGD with Momentum remains highly sensitive to staleness. Notably, with the learning rate tuning, RMSProp no longer diverges, but is actually more robust to staleness than Adam and SGD with Momentum. We have updated manuscript to reflect this new observation. While detailed study of hyperparameter settings is beyond the scope of our work, we will open source our code upon acceptance to make the future reproducibility efforts easier and facilitate the use of simulation study alongside distributed experiments.
>
> LSTM is indeed an interesting piece to add. We have added new results on LSTMs in Appendix A.8 -- we vary the number of layers of LSTMs (see Figure 13) and types of SGD algorithms (see Figure 14), and have observed that (1) staleness impacts deeper network variants more than shallower counterparts, which is consistent with our observation in CNNs and DNNs; (2) different algorithms respond to staleness differently, with SGD and Adam more robust to staleness than Momentum and RMSProp.
>
> We thank the reviewer for the careful review of our theoretical contributions. We especially appreciate the helpful comments that draw the connection between the low gradient coherence at the early phase of optimization and the annealing of the number of workers. Indeed, the convergence analysis of [1] requires the number of parallel workers to follow a \sqrt{K} schedule, where K is the number of iterations. Our work addresses the convergence of non-convex, non-synchronous optimization from a very different starting point than [1] by using gradient coherence, and it seems that similar challenges remains at the initial phase of optimization. We have included a discussion of this connection in the revised manuscript.
>
> [1] Xiangru Lian and et al. Asynchronous parallel stochastic gradient for nonconvex optimization. In NIPS, 2015.

---

> ### Author Response · Authors · 2018-11-29
> **Additional Experiments on LSTM**
>
> LSTM is indeed an interesting piece to add. We have added new results on LSTMs in Appendix A.8 -- we vary the number of layers of LSTMs (see Figure 13) and types of SGD algorithms (see Figure 14), and have observed that (1) staleness impacts deeper network variants more than shallower counterparts, which is consistent with our observation in CNNs and DNNs; (2) different algorithms respond to staleness differently, with SGD and Adam more robust to staleness than Momentum and RMSProp.

---

### Official Review · AnonReviewer2 · 2018-11-05
**The paper addresses asynchronous optimization with a focus on staleness effect. A strong hypothesis is made on the path followed by the optimization walk and concerns should be raised with the hyperparameters in the empirical validation.**

**Rating:** 4
**Confidence:** 5

**Review:**

The papers addresses the important issue with asynchronous SGD: stale gradients.

Convergence is proven under an assumption on the path followed by the optimization walk. Namely, gradient are assumed to be all pointing to the close directions along the walk. My major concern is that this is a strong (if not completely wrong) hypothesis in the practical case of deep learning, with high dimensional models and totally non-convex loss functions (see e.g.
Choromanska et al. 2014).

The paper illustrates empirically the convergence claims, but only under fixed hyper-parameters, which completely illustrates the recent concerns about the reproducibility crisis in ML.

---

> ### Author Response · Authors · 2018-11-19
> **Response to AnonReviewer2**
>
> We thank the reviewer for the comments. Our goals in this work are threefold: (1) Through systematic experiments, we explicitly observe staleness and its impact, for the first time to our knowledge, on 12 key models and algorithms. (2) We introduce gradient coherence (GC), which is related to the impact of staleness for gradient-based optimization. GC can be evaluated in real time during the course of convergence, with minimal overhead, and may be used by practitioners to control delays in the system. (3) Based on GC, we provide a convergence analysis of SGD in non-convex optimization under staleness. With such a broad scope, there is inevitably areas for improvements. We hope that the reviewer will consider the contributions towards making distributed ML more robust under non-synchronous execution as we address the comments:
>
> Regarding the reviewer’s first comment, we would like to clarify that our Definition 1 *does not* require all the gradients to point to close directions along the optimization path. Instead, it only requires the gradients to be positively correlated over a small number of iterations s, which is often very small (e.g. <10 in our experiments). Therefore, Definition 1 is not a global requirement on optimization path. We have clarified this Definition 1 in the revision.
>
> We want to point out that our own results and a number of recent studies show strong evidences that SGD in practical neural network training encourage positive gradient coherence, e.g., Fig. 4(a)(b), and Fig. 5 in our manuscript, [1] and [3], etc. In particular, [1] shows that the optimization trajectories of SGD and Adam are generally smooth, which is also observed in [3] (e.g., Fig. 4 in [3]). These findings suggest that the direction of the optimization trajectory changes slowly during convergence and therefore justifies our Definition 1, even if the gradient direction may oscillate globally [3]. Such findings are perhaps not surprising, because the loss surface of shallow networks and deep networks with skip connections are dominated by large, flat, nearly convex attractors around the critical points [1][2]. This indicates that the degree of non-convexity is mild around critical points. With small batch sizes (32) and skip connections for deep networks in our experiments, our observation of gradient coherence is therefore consistent with the experimental evidence in existing literature.
>
> Regarding the reference (Choromanska et al. 2014) mentioned by the reviewer, even though it shows the (layer-wise) structure of critical points in simple networks with one hidden layer, the more recent works, including those highlighted above, have revealed additional curvature information around critical points and the optimization dynamics for many complex networks. We therefore sincerely ask the reviewer to reevaluate our work in light of these empirical evidence that are consistent with our findings. As pointed out by Reviewer 3, a similar assumption has been made in [4]. We have included these references and discussion in our latest revision.
>
> Regarding to fixed hyperparameters: we have redone all experiments in Fig. 2 with hyperparameter search over the learning rate. We observe the same overall pattern as before: staleness slows down convergence, sometimes quite significantly at high levels of staleness. Furthermore, different algorithms have different sensitivity to staleness, and show similar trends as observed before. For example, SGD with Momentum remains highly sensitive to staleness. Notably, with the learning rate tuning, RMSProp no longer diverges, but is actually more robust to staleness than Adam and SGD with Momentum. We have updated manuscript to reflect this new observation.
>
> Finally, we fully understand the reviewer’s concern about reproducibility. We believe that our simulation work provides a well-controlled environment for future research of distributed machine learning systems. To make the future reproducibility efforts easier and facilitate the use of simulation study alongside distributed experiments, we will open source our code upon acceptance.
>
>
> [1] Li et al. Visualizing the loss landscape of neural nets. To appear in NIPS 2018
> [2] Nitish Shirish Keskar et al. On large-batch training for deep learning: Generalization gap and sharp minima. In ICLR, 2017.
> [3] Eliana Lorch. Visualizing deep network training trajectories with pca. In ICML Workshop on Visualization for Deep Learning, 2016.
> [4] Huo and et al. Training Neural Networks Using Features Replay. To appear in NIPS 2018.

---

### Author Response · Authors · 2018-11-19
**Revision Summary**

We thank all the reviewers for giving valuable feedback to this paper. We have revised the manuscript to incorporate the suggestions from the comments.

We highlight the following revisions:
- We have provided additional discussion and references to recent works presenting empirical evidence consistent with our assumption for Theorem 1.
- We have redone experiments in Fig. 2 with hyperparameter tuning and updated the writing accordingly.
- We have included a brief discussion on how Theorem 1 relates model complexity to the larger slowdown from staleness observed in our experiments.
- We have included reference to [1] which uses the sufficient direction assumption that shares the resemblance to our Definition 1 but differs in certain key aspects.
- We have made further clarifications throughout the manuscript based on reviewers’ comments.
- We have added new results on LSTMs in Appendix A.8 -- we vary the number of layers of LSTMs (see Figure 13) and types of SGD algorithms (see Figure 14) and see how staleness impacts the convergence.

[1] Huo and et al. Training Neural Networks Using Features Replay. To appear in NIPS 2018.

---

### Meta-Review · Area_Chair1 · 2018-12-12
**Good-quality paper**

**Confidence:** 5
**Recommendation:** Accept (Poster)

**Metareview:**

The reviewers that provided extensive and technically well-justified reviews agreed that the paper is of high quality. The authors are encouraged to make sure all concerns of these reviewers are properly addressed in the paper.